# Evaluation of YouTube videos for patients' education on periradicular surgery

**Ahmed Jamleh[1,2]ᵒ\*, Mohannad Nassar[3]ᵒ, Hamad Alissa[1,2], Abdulmohsen Alfadley[1,2]**

**1** Restorative and Prosthetic Dental Sciences, College of Dentistry, King Saud bin Abdulaziz University for Health Sciences, National Guard Health Affairs, Riyadh, Kingdom of Saudi Arabia, **2** King Abdullah International Medical Research Centre, National Guard Health Affairs, Riyadh, Kingdom of Saudi Arabia, **3** Department of Preventive and Restorative Dentistry, College of Dental Medicine, University of Sharjah, Sharjah, United Arab Emirates

ᵒ These authors contributed equally to this work.
\* aojamleh@gmail.com

**Data Availability Statement:** All relevant data are within the paper and its Supporting Information files.

**Funding:** AJ received a research grant (NRC21/012/01/R) from King Abdullah International

## Abstract

The aim of this study was to evaluate the content of periradicular surgery-related YouTube videos available for patients' education. YouTube search was made for videos related to periradicular surgery using specific terms. After exclusions, 42 videos were selected, viewed and assessed by two independent observers. The videos were assessed in terms of duration, days since upload, country of upload, number of views, likes and dislikes, authorship source, viewing rate and interaction index. To grade the content of videos about periradicular surgery, a usefulness score was created with 10 elements based mainly on the American Association of Endodontists guidelines. Each element was given a score of 0 or 1. SPSS software (SPSS Inc, Chicago, IL, USA) was used to analyze data at a 95% confidence level. An inter-evaluator reliability analysis for the scoring system was performed using the Kappa statistic. The videos received an average of 35103.9 views (range: 9–652378) with an average duration of 338.71 seconds (range: 42–2081), respectively. Most videos were provided by individuals (57%). Half of the videos were posted by authors from the United States. The inter-evaluator reliability for usefulness scoring was 94.5%. No video covered the 10 scoring elements completely, presenting very low usefulness scores (mean: 3.2; range: 1–7). The most discussed elements were supporting media (100%) and steps of the procedure (90.5%) followed by indications and contraindications (45.2%) and symptoms (31%). None of the included videos discussed the procedure's cost or prognosis. In terms of usefulness score, no significant difference was detected between different sources of upload (chi-square test, $P > 0.05$). Information on periradicular surgery in YouTube videos is not comprehensive and patients should not rely on YouTube as the only source of information. Dental professionals should enrich the content of YouTube with good quality videos by providing full and evidence-based information that will positively affect patients' attitudes and satisfaction.

Medical Research Center, National Guard Health Affairs, Riyadh, Kingdom of Saudi Arabia. The funder had no role in study design, data collection and analysis, decision to publish, or preparation of the manuscript.

**Competing interests:** No authors have competing interests.

## Introduction

Periradicular surgery is an integral part of endodontic treatment of a tooth with a periradicular lesion which cannot be healed by a traditional root canal treatment or retreatment. This approach is usually perceived as a last resort to retain teeth with persistent periapical lesions despite previous conventional endodontic treatments [1]. The practice of periradicular surgery has increased steadily worldwide; for instance the number of apical surgeries performed in the general dental service in the UK has almost doubled over the past 20 years [2]. The accuracy of this procedure has sharply increased after the adoption of dental operating microscope and other microsurgical tools which enhance the success rate [3], thus making it a more predictable treatment approach and more accepted treatment option [4]. A future increase in the demand for periradicular surgery is anticipated as graduate students and future generation are expected to get sufficient training so as to incorporate it in everyday practice [5].

The need to provide effective and proper patient education regarding periradicular surgery cannot be stressed enough. Providing information to patients in oral health care settings assists them in considering treatment options, increasing knowledge about the procedure and reducing procedure-related distress [6]. It was reported that some patients have difficulties reading endodontic educational materials because of the use of jargons and terminologies that might be confusing to them and those within control of patient education are encouraged to review the readability of their products for their target audience [7]. In clinical practice, information and education about treatment are given verbally or presented in written or audio-visual form. The use of brief informational videos was reported as an efficacious method to provide dental related information and increase knowledge of endodontic patients [6].

The accessibility of smart technologies has had a pronounced influence on the delivery of patient's education in the medical and dental fields. The number of people turning to the internet to search for health related matters continues to grow; it is estimated that one of three US adults uses the internet to diagnose or learn about a health concern [8]. However, the Web should not be considered a substitute for using more reliable health information sources. This trend has been shown to have an effect on the patient healthcare-provider professional relationship [9]. For health minded patients or anxious individuals, the internet is either used to gather information before the appointment in preparation for the visit or it may have even promoted the visit, meanwhile sicker individuals use the internet after the appointment to assist in processing health information that were provided to them [10]. One of the most commonly visited websites which has emerged as a leading video sharing platform for this purpose is the YouTube which received this weight of popularity due to several factors such as ease and convenience of use where unregistered users can watch unlimited numbers of videos [11]. Moreover, no formal identification is needed and it does not have rigid regulations thus it almost allows anyone to publish contents and classify the content into different categories including education and science [12]. Unfortunately, most of the uploaded materials are not peer-reviewed and can be of a variable quality which prompted healthcare professionals to investigate the nature and quality of information available on this website [13,14]. The risk of depending on YouTube as a source for conventional root canal treatment information was highlighted previously; the authors stated the need for endodontic professionals to get in on that act in enjoining health information seekers towards other trusted sources [15]. Healthcare professionals, academic institutions, and professional organizations should share this responsibility of refining the content of this site directing patients to reliable information sources [13].

Up to our knowledge, there is a paucity of scholarly literature in the content of YouTube videos on periradicular surgery. In this study we aimed to evaluate the content, accuracy and

quality of the most viewed periradicular surgery-related videos available on the aforementioned website.

## Material and methods

### YouTube search strategy

On the fifth of February 2021, YouTube search was made for videos related to periradicular surgery. The following related terms were used: (1) apical curettage; (2) root end resection; (3) root end surgery; (4) root end filling material; (5) periradicular surgery; (6) periapical surgery; (7) apicoectomy; (8) retrograde root canal treatment; (9) retrograde endodontics; (10) retrograde filling material; (11) endodontic surgery; (12) microsurgical endodontics; (13) endodontic microsurgery; (14) surgical endodontics; and (15) non-conventional endodontic therapy.

Past researches proposed that the majority of YouTube users scan the first 30 videos thousand times per day [16]; therefore, we analyzed the first 30 videos for each search term. The search was conducted using an incognito window with a cache clean and unlogged browser to prevent robot learning and under default settings without any filters for sorting by relevance.

### Selection of videos

Initial screening was performed to include videos related to periradicular surgery. A video was excluded if it had one the following criteria:

- videos lacking audio or visuals

- videos about other types of endodontic treatments

- videos describing the findings of a research paper/project

- videos primarily directed to a specialized audience (i.e., a conference/scientific meeting presentation or a medical/dental school lecture)

- non-English language videos

- advertisements, stories or songs

- drama-based or satirical videos

An account on YouTube was created for the purpose of the study and the included videos were stored following the removal of duplications.

### Evaluation of videos

The videos were entirely viewed and analyzed independently by two observers, who are endodontists, to get information about: (1) video's duration, (2) date of upload, (3) country of upload, (4) numbers of views and (5) number of likes and dislikes. The upload source was identified and categorized based on authorship.

To analyze and grade the value of videos in providing the viewers adequate information about periradicular surgery, a usefulness score was created based mainly on the American association of endodontists (AAE) guidelines. The scores were in the range of 0–10: A score of 10 reflected that the video touched on the following aspects of periradicular surgery: definition, etiology, symptoms, indications and contraindications, steps of the procedure, cost, postoperative care, complications, prognosis and supporting media (images and videos) (Table 1). Each element was given a score of 0 or 1 based on its consistency with the information usefulness about endodontic surgery. Based on this, video content was deemed poor (0–3 points), moderate (4–7 points), or good (8–10 points).

**Table 1. A usefulness scoring system and observation rate for videos about "periradicular surgery".**

| Scoring element | Score | Observation rate |
|---|---|---|
| Definition | 1 | 19% |
| Etiology | 1 | 14.3% |
| Symptoms | 1 | 31.0% |
| Indications and contraindications | 1 | 45.2% |
| Steps of the procedure | 1 | 90.5% |
| Cost | 1 | 0.0% |
| Postoperative care | 1 | 4.8% |
| Complications | 1 | 14.3% |
| Prognosis | 1 | 0.0% |
| Supporting media (videos and images) | 1 | 100.0% |
| TOTAL | 10 | - |

The interactions of users with the included videos were assessed based on the interaction index and the viewing rate by using the formulae: [(number of likes-number of dislikes/ total number of views) * 100%)] and [(number of views/ number of days since upload) * 100%)], respectively.

## Statistical analysis

An inter-observer reliability analysis for the usefulness scoring was conducted using the Kappa statistic to determine the variability. In the different opinion event, a consensus was obtained after reviewing the related videos.

Statistical tests were run using SPSS software (SPSS Inc, Chicago, IL, USA) to investigate the relationship between content usefulness and video characteristics and demographics. Continuous variables were studied using the Kruskal-Wallis test and Mann-Whitney U test. Categorical variables were studied using the chi-square test. The significance level was set at 5%.

## Results

The content usefulness was determined using 10 elements (Table 1). The most covered elements were supporting media (100%) followed by steps of the procedure (90%), indications and contraindications (45.2%) and symptoms (31%). Less than 20% of the videos discussed definition, etiology, complications or postoperative pain. None of the videos discussed the procedure's cost or prognosis (Table 1).

Video characteristics are provided in Table 2. Half of the videos were from the United States and the rest were from India, the United Kingdom, Canada, Greece, Jordan, Brazil, Italy, South Korea, Spain, and Germany. The source of authorship was identified as individuals (57.1%), companies (31%) or academic institutions (11.9%).

Descriptive statistics of evaluated videos are provided in Table 3. The included videos have an average of 35103.9 (Range: 9–652378) views with an average duration of 338.7 sec (Range: 42–2081 sec). The mean interaction index score was 1.0 (Range: 0–7.6). The mean number of "likes" was 126.3 (Range: 0–2100). The mean usefulness score was 3.2±1.4 with a range of 1–7. In terms of content usefulness score, 61.9% of the videos were classified as poor, 38.1% as moderate, and 0% as good. The moderate videos were longer in duration with higher number of views, number of likes, interaction index and viewing rate when compared with poor videos (p< 0.05) (Table 4).

No statistically significant difference was found between video source of authorship and content usefulness score (p>.05). No difference was found between the authors in terms of numbers of views, number of likes, number of dislikes, interaction index, viewing rate or

**Table 2. Video characteristics.**

| | | N | % |
|---|---|---|---|
| **Country** | The United States | 21 | 50 |
| | India | 7 | 16.7 |
| | The United Kingdom | 4 | 9.5 |
| | Canada | 2 | 4.8 |
| | Greece | 2 | 4.8 |
| | Jordan | 1 | 2.4 |
| | Brazil | 1 | 2.4 |
| | Italy | 1 | 2.4 |
| | South Korea | 1 | 2.4 |
| | Spain | 1 | 2.4 |
| | Germany | 1 | 2.5 |
| **Source of authorship** | Individual | 24 | 57.1 |
| | Company | 13 | 31.0 |
| | Academic | 5 | 11.9 |
| **Content usefulness score** | Good | 0 | 0 |
| | Moderate | 16 | 38.1 |
| | Poor | 26 | 61.9 |

**Table 3. Descriptive statistics of evaluated videos (n = 42).**

| Demographics | Mean±SD | Median | Q1-Q3 | Min-Max |
|---|---|---|---|---|
| Video length (in seconds) | 338.7±361.3 | 241.5 | 136.5–371 | 42.0–2081.0 |
| Days since upload | 2002.5±1243.4 | 1764 | 869–3400 | 133–4242 |
| Numbers of views | 35103.9±104231.5 | 5590.5 | 645.8–22675 | 9–652378 |
| Number of likes | 126.3±359.2 | 24 | 6.3–9. | 0.0–2100.0 |
| Number of dislikes | 12±34.9 | 1 | 0–6 | 0.0–203.0 |
| Interaction index | 1.0±1.6 | 0.4 | 0.1–1.0 | 0.0–7.6 |
| Viewing rate | 2073.1±6402.1 | 329.2 | 62.7–1099.9 | 6.8–33335.6 |
| Content usefulness score | 3.2±1.4 | 3 | 2–4 | 1–7 |

**Table 4. Comparison of YouTube video demographics based on the usefulness score categories.**

| Demographics | Poor (n = 26) | | Moderate (n = 16) | | Good (n = 0) | | P- value[a] |
|---|---|---|---|---|---|---|---|
| | Mean±SD | Median | Mean±SD | Median | Mean±SD | Median | |
| Video length (in seconds) | 253.5±241.7 | 185 | 477.3±475.7 | 343 | 0 | 0 | 0.017 |
| Days since upload | 1764.5±1224.7 | 1629.5 | 2389.3±1211.7 | 2543.5 | 0 | 0 | 0.087 |
| Numbers of views | 8102.3±10045.7 | 2941.5 | 78981.4±161894.9 | 18554 | 0 | 0 | 0.01 |
| Number of likes | 36.9±54.5 | 13.5 | 271.4±558.0 | 31.5 | 0 | 0 | 0.015 |
| Number of dislikes | 2.9±4.6 | 1 | 26.7±54.1 | 1.5 | 0 | 0 | 0.234 |
| Interaction index | 457.2±632.2 | 165.9 | 4699.1±9974.2 | 684.2 | 0 | 0 | 0.013 |
| Viewing rate | 36.9±54.5 | 13.5 | 271.4±558.0 | 31.5 | 0 | 0 | 0.015 |

content usefulness (p>0.05). The longest duration was found in videos uploaded by academic institutions (792.2±780.6 sec), followed by individuals (356.5±231.5 sec) and companies (131.5 ±85.2 sec) (p<0.001). Perfect agreement between the observers in scoring the videos was detected (Kappa = 0.945).

## Discussion

Past experiments have studied the YouTube videos contents for various oral-health-related topics [15,17–25]. In the present study, the video content was evaluated based on the AAE guidelines regarding periradicular surgery. Excellent agreement was found between the observers. Patient satisfaction is related to the amount of information received before procedure mainly regarding the steps and complications [18]; in the present study, none of the videos addressed all the elements needed to fully understand the procedure from a patient perspective.

All the evaluated videos were supported with images and audio which could seemingly facilitate knowledge acquisition, thus making them more frequently watched. However, the usefulness score of two-thirds of the tested videos was poor whilst none was qualified to be good. These findings were due to the fact that most of the videos lacked definition, etiology, complications and postoperative pain aspects, and none of them discussed the procedure's cost or prognosis. None of the videos covered all the elements required to understand the procedure which resulted in an average usefulness score of 3.2±1.4 out of 10. Hence, YouTube is not a suitable single source to obtain comprehensive information about periradicular surgery. Inadequate scores of YouTube videos discussing other dental-related topics were also previously reported [15,18–21,24]. It is known that YouTube videos are not peer-reviewed nor based on scientific evidence [13,14] which adds a question mark over the reliability and quality of the information being presented. In the current study, most of the videos were uploaded by individuals (57.1%), a factor that we believe has contributed to a low usefulness score. In our opinion, this deficiency can be tackled by the involvement of foremost dental academic institutions and organizations in producing high quality YouTube videos. Indeed, YouTube studies have highlighted the importance of professionals' contribution to high-quality videos as relevant patient-information sources [26,27].

Although low number of the included videos were found in YouTube, it is expected in the near future to have more videos with the increased demand and training sessions for it among postgraduate students [5].

Considering the anticipated increase in demand for periradicular surgery [5], comprehensive patient education materials is highly needed to reduce procedure-related distress [6]. Patients tend to search for health related matters through YouTube platform since it provides health information for free with audiovisual materials. A previous study reported the risk behind patients' reliance on YouTube as a source of information about non-surgical root canal treatment [15]. Fortunately, the feasibility of periradicular surgery as a treatment option is firstly recognized by the patients after receiving detailed clinical and radiographic examinations and is usually suggested by the dental professional who should discuss these aspects in the clinic and direct patients to high quality sources to obtain verified and up-to-date information, thus minimizing the amount of misleading information and avoiding videos with limited usefulness.

Based on the usefulness scoring, moderate videos showed favorable demographic data. It is noted that moderate videos received higher number of views whilst poor videos received as low as 9 views. Moreover, videos with higher usefulness scores had longer duration with higher number of views, number of likes, interaction index, viewing rate and content usefulness. These findings are inconsistent with some previously published studies [11,13,18,25,28,29]. However, other studies indicated that videos with high usefulness scores were longer [20–23]. In this study, no significant differences were found in video demographics among the different authorship sources except for video duration where academic institutions uploaded longer videos which also gave better usefulness scores. The latter might be ascribed to the ability of

covering many aspects of the treatment in a sufficient way with longer videos [25]. The use of videos for oral health education is an enjoyable and preferred method by patients, however, the fun in learning was previously reported to be more with patients who watched the video from 5 to 20 minutes [30]. This calls for the need of studies evaluating the effect of video length on the usefulness of the presented materials and the engagement of the patient.

YouTube's features such as "like" and "dislike" is a way that indicates the viewers' feedback and engagement. It can also help dental professionals to measure the video usefulness and patients' satisfaction. It is noteworthy to mention here, that we found a higher number of likes in the videos that fall in the moderate usefulness category. Another way to measure viewers' interactions with videos is by assessing the interaction index and viewing rate [13,18,21]. Previous YouTube studies reported varying results about the viewers' interaction [11,13,26,28,29,31,32]. In our study, the interaction index and viewing rate were better in videos with higher scores. By default, YouTube arranges videos based on number of views, interaction index and viewing rate [15]. Based on this, useful videos will be chosen and watched. Also, videos can be sorted based on number of views, days since upload, viewing rate and video duration.

The health professionals should inform their patients about misleading and unreliable information on YouTube and recommend alternative reliable online sources. Social media, including YouTube, are used by health professionals, mainly for marketing and advertising purposes [33,34]. A previous study showed that that only 4.2% of the dental educators uploaded informative videos on YouTube [35]. Therefore, it is crucial to encourage healthcare professionals to upload videos within the context of healthcare's good practice and ethical rules.

As with other studies, there are a few limitations in the current study. Our results only reflected information available in English which were collected at the time of the search. It is notable that YouTube accepts videos with a wide range of languages and is being updated continuously by deleting and uploading videos without scientific and quality checks. It was also impossible to figure out the audience nature. Moreover, patients might use other search terms which can eventually show different results. To make this negligible, efforts were made to retrieve a wide range of possible YouTube videos related to periradicular surgery wherein 15 search terms were used in this study. Lastly, the usefulness scoring was made by endodontists wherein different results could have been attained if the scoring had been conducted by other healthcare professionals or patients. Nevertheless, excellent agreement was found between the two observers who come from a specialization background that is directly related to the periradicular surgery treatment modality.

## Conclusion

There are a small number of videos on YouTube with adequate information on periradicular surgery. Since the majority of videos received low scores, patients should not rely on YouTube as the only source of information on periradicular surgery. Dental professionals should enrich the content of YouTube with good quality videos by providing comprehensive and evidence based information which can affect patients' attitudes and satisfaction.

## Supporting information

**S1 Table. Data collection sheet for YouTube videos about periradicular surgery.**
(XLSX)

## Author Contributions

**Conceptualization:** Ahmed Jamleh, Mohannad Nassar.

**Data curation:** Ahmed Jamleh, Mohannad Nassar.

**Formal analysis:** Ahmed Jamleh, Mohannad Nassar.

**Investigation:** Ahmed Jamleh, Mohannad Nassar, Hamad Alissa, Abdulmohsen Alfadley.

**Methodology:** Ahmed Jamleh, Mohannad Nassar, Hamad Alissa.

**Project administration:** Ahmed Jamleh.

**Supervision:** Ahmed Jamleh, Mohannad Nassar.

**Visualization:** Ahmed Jamleh.

**Writing – original draft:** Ahmed Jamleh, Mohannad Nassar, Abdulmohsen Alfadley.

**Writing – review & editing:** Ahmed Jamleh, Mohannad Nassar, Abdulmohsen Alfadley.

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
