## [Decision Letter · Decision Letter 0]

16 Nov 2021

PONE-D-21-31439Evaluation of YouTube Videos for Patients' Education on Periradicular SurgeryPLOS ONE

Dear Dr. Ahmed Jamleh,

Thank you for submitting your manuscript to PLOS ONE. After careful consideration, we feel that it has merit but does not fully meet PLOS ONE’s publication criteria as it currently stands. Therefore, we invite you to submit a revised version of the manuscript that addresses the points raised during the review process.

Please respond to the comment mentioned in point 5. Review comments to the Author.

We look forward to receiving your revised manuscript.

Kind regards,

Tanay Chaubal

Academic Editor

PLOS ONE

Journal Requirements:

[Authors of this study deny any conflict of interest. This work was supported by a research grant (NRC21/012/01/R) from King Abdullah International Medical Research Center, National Guard Health Affairs, Riyadh, Kingdom of Saudi Arabia.]

 [No]

Additional Editor Comments:

Dear Ahmed Jamleh,

Please respond to the comments made by the reviewer.

Thank you.

Reviewers' comments:

Reviewer's Responses to Questions

**Comments to the Author**

1. Is the manuscript technically sound, and do the data support the conclusions?

Reviewer #1: Partly

2. Has the statistical analysis been performed appropriately and rigorously? 

Reviewer #1: Yes

3. Have the authors made all data underlying the findings in their manuscript fully available?

Reviewer #1: Yes

4. Is the manuscript presented in an intelligible fashion and written in standard English?

Reviewer #1: Yes

5. Review Comments to the Author

Reviewer #1: how has the appearance video in concern taken into account on social media ? what do u mean by the videos from country of origin? is there any countries upload restricted in your country? can u please let us know the rationale of your study?

6. PLOS authors have the option to publish the peer review history of their article (what does this mean?). If published, this will include your full peer review and any attached files.

Reviewer #1: No

---

## [Author Response · Author response to Decision Letter 0]

18 Nov 2021

Tanay Chaubal

Academic Editor

PLOS ONE

Dear Dr. Tanay;

It is an immense pleasure to submit our revised manuscript entitled: “Evaluation of YouTube videos for patients' education on periradicular surgery”. The authors ensured that the manuscript meets PLOS ONE's style requirements and have made changes to the manuscript according to the suggestions from the editor and the reviewers; the changes follow and are made in the revised manuscript using “Track Changes” mode.

The funding information was removed from the “Acknowledgment” section which has to appear in the “Financial Disclosure”.

Reference list was updated and formatted as requested.

We are grateful for the attention and effort spent in reviewing our work, and valuable comments made by the respectful editor and reviewers.

We sincerely hope that the revised manuscript is now suitable for publication in the PLOS ONE Journal.

Sincerely yours,

Reply to reviewer

How has the appearance video in concern taken into account on social media? What do u mean by the videos from country of origin? Is there any countries upload restricted in your country? Can u please let us know the rationale of your study?

Periapical surgery is increasingly becoming a common treatment option in the field of endodontics, and more clinicians including postgraduate students are being trained on this treatment modality. Due to the expected higher demand, the need for good quality educational materials on periapical surgery cannot be stressed enough. With the growing number of patients using social media, it appears that there is a need to evaluate the usefulness of periapical surgery-related videos posted on social media as a source of patient education. One of the most commonly used platforms for this purpose is YouTube due to the several advantages it offers over other platforms, and it is slowly becoming one of the primary sources for patients seeking specific health-related information. We conducted this study to analyze YouTube as a source of patient educational material on periapical surgery. The findings of our study highlight an immediate need for higher quality educational videos on periapical surgery as the existing videos content and quality vary significantly and none of the evaluated videos covered all the essential components related to this type of surgery. This responsibility should be shared among dental organizations, academic and professional clinicians.

The country of origin (country of upload) indicates which countries are talking about this issue. In this study, half of the videos were from the United States.

The search was conducted using an incognito window with a cache clean and unlogged browser to prevent robot learning and under default settings without any filters for sorting by relevance. However, some YouTube videos may not be available in the search as the video owners have chosen to make their content available only to certain countries/regions.

---

## [Editor Report · Decision Letter 1]

1 Dec 2021

Evaluation of YouTube videos for patients' education on periradicular surgery

PONE-D-21-31439R1

Dear Dr. Ahmed Jamleh,

We’re pleased to inform you that your manuscript has been judged scientifically suitable for publication and will be formally accepted for publication once it meets all outstanding technical requirements.

Kind regards,

Tanay Chaubal

Academic Editor

PLOS ONE

Additional Editor Comments (optional):

Dear Dr Ahmed Jamleh,

I am pleased to accept your manuscript for publication.

Thank you.
---

## [Editor Report · Acceptance letter]

3 Dec 2021

PONE-D-21-31439R1 

Evaluation of YouTube videos for patients' education on periradicular surgery 

Dear Dr. Jamleh:

I'm pleased to inform you that your manuscript has been deemed suitable for publication in PLOS ONE. Congratulations! Your manuscript is now with our production department. 

Kind regards, 

on behalf of

Dr. Tanay Chaubal 

Academic Editor

PLOS ONE